# Transdisciplinary Scientific Strategies for Soft Computing Development: Towards an Era of Data and Business Analytics

**Rafael A. Espin-Andrade** [1], **Witold Pedrycz** [2], **Efrain Solares** [1,*] and **Laura Cruz-Reyes** [3]

1 Faculty of Accounting and Management, Universidad Autónoma de Coahuila, Torreon 27000, Mexico; rafaelalejandroespinandrade@gmail.com
2 Department of Electrical and Computer Engineering, University of Alberta, Edmonton, AB T6G 2R3, Canada; wpedrycz@ualberta.ca
3 Division of Graduate Studies and Research, Instituto Tecnológico de Ciudad Madero, Tampico 89440, Mexico; lauracruzreyes@itcm.edu.mx
* Correspondence: efrain.solares@uadec.edu.mx

**Abstract:** This research is a review and analysis paper that offers a transdisciplinary, methodological, and strategic vision for soft computing development towards a wider favorable impact in data analytics. Strategies are defined, explained, and illustrated by examples. The paper also shows how these strategies are expressed in three dimensions of an ambitious actions plan. They are all integrated into a master strategy called wide knowledge discovery, which offers a way towards the augmented analytics paradigm. Some contributions of this work are defining what kind of mathematical elements should be introduced into soft computing towards a better impact on the area of data analytics, offering orientation towards building new mathematical elements, and defining why and how they can be introduced.

**Keywords:** fuzzy logic; data analytics; soft computing; computational intelligence; knowledge discovery

**MSC:** 68T30

## 1. Introduction

An analysis of how computational intelligence (considered a synonym of soft computing) should be transformed to get effective new tools of data and business analytics is an excellent pluralist approach that offers orientation towards the construction of new mathematical elements, useful to enhance the data-oriented analysis and applications. That is the objective of this paper as well as its main contribution.

Mathematical thinking as relevant component of science, is about theories and applications and creating new theoretical elements towards problem solutions. Soft computing is an interesting multidisciplinary space where the mathematical compass has been more critical towards the broad spectrum of problems treated into them. The interaction among engineers, social scientists, computer scientists, and mathematicians in that open community is very much oriented to applied sciences, and that objective-oriented way to think is especially useful and used in this area.

That is a powerful reason why a paper like this is important for a journal oriented to mathematics, especially in the context of the special issue "Soft computing: Theories and Applications II".

Analytical thinking used in data analytics and business analytics is one of the most important competencies demanded by the so-called Education 4.0 (E4.0) part of the emergent paradigm of the new era, Industry 4.0 (I4.0): The fourth industrial revolution.

E4.0 and I4.0 multiply the importance and complexity of intelligent systems that provide solutions to decision making problems. Some techniques used by such systems are

a permanent connection to the cloud, processing of growing amounts of data, connection to multiple services and production processes, interconnections with networks of physical, digital, and biological elements, and relevant participation of human beings.

Computer enhancement of human intelligence and human capabilities for problem-solving and decision-making has been called augmented intelligence, and more recently, as hybrid augmented intelligence. The latter explicitly emphasizes the two elements or strategies of this enhancement: (1) using Artificial Intelligence for search automatization and natural language communication from one side; (2) embedding a cognitive model into the machine learning system [1–4].

The data analytics paradigm has evolved towards the active use of augmented intelligence, which has been called augmented analytics [5–8]. The former allows the collaboration between humans and computers through natural language and uses analytics to solve problems of interest interactively. The application of the second strategy should allow advances toward: two aspects: (1) More direct orientation to decision-making tools; (2) Better participation of experts and decision makers in the analytics process, by the suggestion of hypotheses, concepts, and decision-making alternatives.

It has developed well-known tools of business-oriented analytics to automatically solve standard problems of data mining associated with diverse decision-making problems. These tools are used to present a synthesis of the information with excellent capabilities for visualization and the notable use of dashboards and offer possibilities of interaction (even to the point of using natural language consulting) with the experts and decision-makers [4].

Eurekas Community (www.eurekascommunity.org, accessed on 30 April 2021) is an international scientific community working with data analytics and decision making, emphasizing interdisciplinary and transdisciplinary approaches. Wide knowledge discovery (WKD) is a transdisciplinary scientific strategy developed by Eurekas to contribute to the use and development of Business Intelligence and Business Analytics technology. It joins knowledge engineering, decision analysis, and knowledge discovery in just a process towards an effective decision-making support [5].

New theoretical developments like compensatory fuzzy logic (CFL) and, more recently, Archimedean compensatory fuzzy logic (ACFL), are transdisciplinary theories with superb properties for interpretability with language [9–12]. A complete description of a three dimensions research program in course can be found in Appendix A.

More directly oriented to decision-making tools and better participation of experts and decision-makers in the analytics process, by the suggestion of hypotheses, concepts, and decision-making alternatives evaluation, are relevant for the successful evolution of Data Analytics towards the augmented analytics (AA) paradigm.

Observe that business analytics is a particular case of data analytics involving business activities and management and economic categories, knowledge discovery is an important activity of data analytics, complemented with other essential activities like decision making analysis, knowledge engineering, and inference.

Inference includes the effective use of the knowledge discovered towards conclusions of the values of certain variables (decision variables) considering the values of other ones (condition variables). Augmented analytics is a new way to do data analytics using augmented intelligence, and hybrid augmented analytics is a way to do augmented analytics, including the embedment of cognitive models. A particular case of inference is forecasting. An important case of decision analysis is planning, which used to be made by optimization.

Strategies are ways to advance towards a described imagined state in the future that use to be called vision of future (VF). A strategy relevant towards it, which integrates a group of them towards the vision, is called Master strategies.

The AA paradigm is the consensual VF accepted for data and business analytics communities; WKD is the master strategy used explicitly by Eurekas Community to advance towards AA.

The paper demonstrates how the presented strategies are included in an ambitious plan of Eurekas Community towards data and business analytics contribution where soft computing has a relevant role.

Section 1 is a strengths, weaknesses, opportunities, and threats (SWOT) Analysis, Section 2 offers philosophical and cognitive arguments behind WKD, Section 3 is devoted to the Strategies used.

Related work and plans are in the fourth section. The Conclusion and Recommendation section is the last one. A list of acronyms can be found in Appendix B.

## 2. SWOT Analysis of Computational Intelligence towards Data and Business Analytics

The emergence of fuzzy sets and fuzzy logic theories is celebrated, commemorating the date of Lofti Zadeh's first paper publication [13]. It was honored 50 years after such publication by some relevant scientists of the areas with an interesting collection of papers in 2015 [14–16]. Fuzzy Sets and Fuzzy Logic theories are considered by wide circles of engineering, mathematics, computer sciences, social sciences, and other scientific areas, the sciences of vagueness, a particular form of uncertainty associated with information by words.

Soft computing, also created by Zadeh, is considered a complement to fuzzy sets and Fuzzy Logic. Soft computing comprises neural networks, evolutionary and other heuristic algorithms, probabilistic thinking, and Hybrid Systems, among others.

Some strengths and weaknesses of Soft computing:

- The Fuzzy disciplines consideration as sciences of vagueness is probably one of the most relevant strengths of Soft Computing.
- The growing role of mathematical fuzzy logic or narrow fuzzy logic as part of soft computing is relevant for the advancing theoretical and successful applications [17–22].
- Certain advances towards natural language interpretability, like computing with words (CWW) [23–28], linguistic data summarization (LDS) [29–32], linguistic integration of membership functions by operators (LIMFO) [31–40], and Mamdani fuzzy systems (MFS) constitute a powerful strength [41–50].
- Hybridization with neural networks and deep learning [51–59] are relevant strengths as well.
- However, those advances have not been enough for meaningful achievements in natural language recognition and processing. This condition is a relevant weakness in comparison with the natural expectations of the first decades of soft computing development and the extraordinary results obtained by other ways like deep learning [14–16].
- Another two important strengths are the graphical interpretability of fuzzy logic and the development of evolutionary algorithms and metaheuristics [14–16,60]. They could be especially important together as ways to get synergically promising approaches for the usual hybridization employed in knowledge discovery: A way of representation, a way of evaluation and a method of searching.
- The two other principal weaknesses of fuzzy systems as part of soft computing are probably the evolution of fuzzy control towards functional adjustment techniques not interpretable by natural languages, like the classical Mamdani systems, which principal limitation have been the lack of accuracy [41–50].
- These two weaknesses combination has produced the myth that interpretability and accuracy are not compatible. That is possible because the traditional concept of Interpretability in fuzzy logic is associated with different measures highly connected with simplicity. The authors of this paper consider that interpretability by natural language, like any interpretation, is practical just because it contributes by the interpretation to efficacy and efficiency of modeling and analysis. Accuracy is one of the essential properties of those two attributes. This treatment of interpretability should be considered another critical weakness. The formal treatment of the concept of interpretability in science is not associated with the possibility to be understood directly

by a person, but it relates to a translation process disconnected with the participation of individuals [61–65].

Some Opportunities and Threats:

- The consensus of the scientific communities of artificial intelligence, machine learning as well as data and business analytics concerning the importance of augmented intelligence and augmented analytics paradigms constitute a relevant opportunity to fuzzy logic and soft computing associated with its relationship with natural language, as a clear need towards comprehensive or integrative computational cognitive models which are not available yet [1–5].
- The existence of the very consensual Chomsky grammar approach of universal transformational generative grammar could make that opportunity even more relevant, giving spaces for the construction of models based on the so-called minimalist program [60,61].
- A growing consensus of the need for transdisciplinary science and pluralism in science and logics because they are ways that benefit soft computing as the source of solutions and the historical emphasis in their hybridization [66–69].
- The extraordinary accuracy results of deep learning in different applications constitute a threat to the use of other models.
- However, the black box characteristic of those models is an opportunity because of the possible creation of hybrid models combining accuracy arising from deep learning and interpretability stemming from fuzzy logic.

Summarizing:

- Strengths: 1. Recognition of fuzzy logic as the science of vagueness (SciV), 2. the growing role of mathematical fuzzy logic (RMFL), 3. advances towards interpretability by natural language (ATNLI), 4. graphical interpretability by trees, graphs, and networks (GIFL), 5. development of evolutionary algorithms and metaheuristics (DEA).
- Weaknesses: 1. Non-appreciable results in natural language treatment from fuzzy sets and fuzzy logic (NARNLT), 2. fuzzy control deviation towards functional adjustment techniques. Limitations of Mamdani fuzzy systems (FCD).
- Opportunities: 1. Transdisciplinary science and scientific and logical pluralism (TS-SLP), 2. lack of general cognitive models (LGCM), 3. lack of interpretability of deep learning (LIDL), 4. existence of universal transformational generative grammar, a consensual scientific linguistic model (TGG).
- Threads: 1. Deep learning extraordinary accuracy results (DLAR).

Table 1 illustrates the strategic cross impacts:

**Table 1.** Cross-impact matrix.

|           | **1 TS-SLP** | **2 LGCM** | **3 LIDL** | **4 TGG** | **5 DLAR** |
|-----------|:------------:|:----------:|:----------:|:---------:|:----------:|
| 1 SciV    |              | x          | x          | x         |            |
| 2. RMFL   | x            |            |            |           | x          |
| 3 ATNLI   |              | x          | x          | x         |            |
| 4 GIFL    | x            |            | x          |           | x          |
| 5 DEA     | x            |            |            |           | x          |
| 6 NARNLT  |              | x          | x          | x         |            |
| 7 FCD     | x            |            |            |           | x          |

The opportunity associated with the acceptance of transdisciplinary science and scientific-logical pluralism (TS-SLP) can be better used, considering the following strengths/weaknesses: growing role of mathematical fuzzy logic (RMFL), Graphical interpretability of fuzzy logic (GIFL), development of evolutionary algorithms (DEA), and fuzzy control deviation (FCD). All those properties of soft computing are associated with improvements possible and necessary by hybridization.

The opportunity lack of general cognitive model should consider the strengths/weaknesses: science of vagueness acceptation for fuzzy disciplines (SciV), advances towards natural language interpretability (ATNLI) and non-appreciable results in natural language treatment from fuzzy logic (NARNLT). They are all related to the natural language use improvement by fuzzy logic.

The opportunity lack of interpretability of deep learning should take into account the following strengths/weaknesses: science of vagueness acceptation for fuzzy disciplines (SciV), advances towards natural language interpretability (ATNLI), graphical interpretability of fuzzy logic and non-appreciable results in natural language treatment from fuzzy logic (NARNLT). All those are related to the interpretability of fuzzy logic.

The opportunity universal transformative grammar should consider the strengths/weaknesses: science of vagueness acceptation for fuzzy disciplines (SciV), Advances towards natural language interpretability (ATNLI) and non-appreciable results in natural language treatment from fuzzy logic (NARNLT). They are all related to the natural language interpretability improvement by fuzzy logic.

The thread deep learning extraordinary accuracy results (DLAR) should consider the following strengths/weaknesses: growing role of mathematical fuzzy logic (RMFL), graphical interpretability of fuzzy logic (GIFL), development of evolutionary algorithms (DEA), and fuzzy control deviation (FCD). All those properties of soft computing are associated with improvements possible and necessary by hybridization.

### 3. Wide Knowledge Discovery Strategy towards AA Paradigm: Philosophical, Cognitive and Strategic Arguments

Analytic philosophy has been an important tendency especially relevant for Philosophy of science and philosophy of language. It is a very diverse way to think principally in the United States, the United Kingdom, Australia, and all the community of English speakers over the world. Analytic Philosophy, developed by Wittgenstein and Bernard Russell among others, contributes to the analysis using language, and especially analysis through an ideal language associated with logics. The latter analysis constitutes a dominant role in Philosophy and sciences. That ideal language, different from ordinary or natural language and different to scientific language as well, should avoid vagueness, ambiguity, and any other uncertainty, allowing for application in philosophy and sciences with a particular way to deal with knowledge, covering all the possibilities of any reality without contradictions [70–73].

Sciences acquire special meaning in these times of the Society and Economy of Knowledge. It is very influential in everyday life and especially relevant for the business activity characterized more and more by strategies associated with high added value products. This is multiplied by the emergence of Industry 4.0 that makes technology, and consequently science, even more relevant. An important manifestation of these changes—multiplying the role of science in society—is the data analytics multiplied relevance, understood as the use of mathematical statistics and artificial intelligence in the exploration of data, looking for patterns and knowledge useful for decision making in business and other spaces of application. A compatible approach is pluralism, a new tendency in science. They study different elements of a common studied object. Trying to understand a discipline from different scientific values is a new way to do, which can make stronger interdisciplinary and transdisciplinary approaches. Bringing back analytic philosophy and other philosophical schools discussions into the framework of the pluralist spirit emerging lastly in science and logic [74,75] is relevant for building a new analysis, and should no longer be an elitist dialogue, but a relevant one with a wide participation of the scientific community.

The clarification intention of Ludwig Wittgenstein (1889–1951) [76,77] and the first pluralist approach of logical positivism by Rudolph Carnap (1891–1970) [78,79] should re-emerge with new usefulness and relevance in the era of artificial intelligence, especially when the dialectic negation of it has created augmented intelligence and augmented analytics, emerging disruptively everywhere.

The new analytics should be consequently, joining scientific values without privileges for specific ones. Philosophical schools' values like the role of experience (phenomenology), the emphasis in results, problem-solving and decision-making (pragmatism), intuition and the spontaneous emergence of the hypotheses as consequence of direct experience (vitalism), as well as experimental emphasis and facts based theoretical elaboration of hypothesis and theories (positivism), should be joined in that new emergent way to face scientific analysis [80,81].

Because of the evolution towards a systematic and non-elitist role of science in life, the language role coming from the classical analytic philosophy should be able to play a main role, now without idealizations, looking for scientifically implementing those values, using natural language and accepting uncertainty and vagueness as substantial part of language, knowledge and everyday life.

Wide knowledge discovery (WKD) is a pluralist strategy, considering all the mentioned values before, using human and factual sources. It departs from the use of representations of general knowledge and posterior contextualization and parametrization from new particular and singular knowledge from human and factual sources, giving a real decision-making orientation of data and business analytics, as it is illustrated by Figure 1.

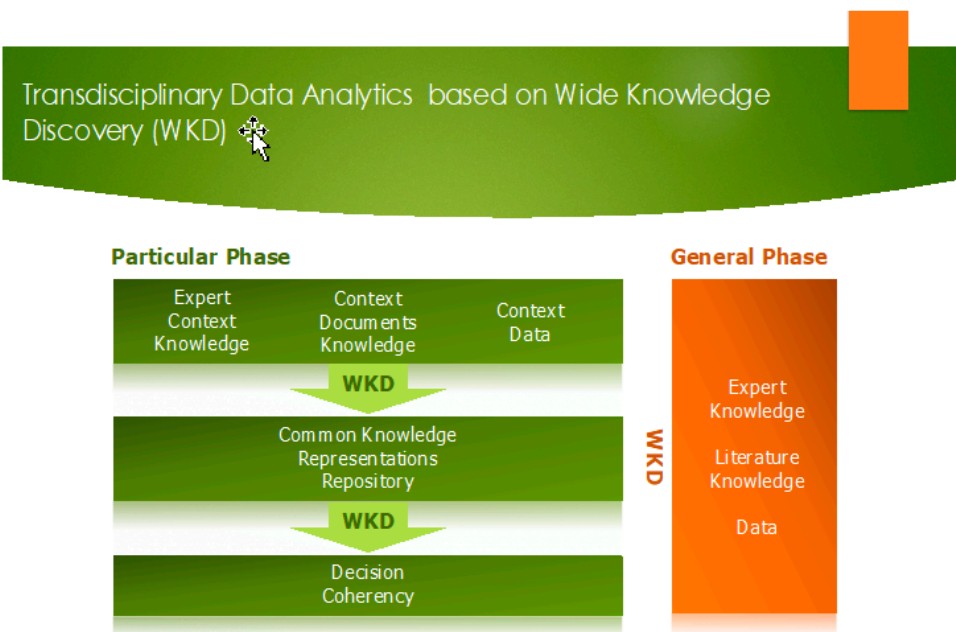

**Figure 1.** Transdisciplinary Data Analytics and WKD.

Wide knowledge discovery should be implemented as hybrid systems of three elements:

A way to represent, a way to evaluate and a way to search. This general wide knowledge discovery, need to hold important requirements:

1.  The way to represent knowledge should be general; then, all the knowledge could be represented. That general way of representation can be transformed easily to natural language and different graphical representations, facilitating the understanding and illustration of the knowledge represented
2.  The way to evaluate knowledge should be a hybrid approach that joins important theoretical elements used in knowledge processing and is represented by natural language and different graphical approaches.
3.  The way to search for knowledge could be in the form of optimization, and particularly, through heuristics associated with different ways of representation.

WKD is a transdisciplinary strategy that proposes integrating two aspects: (a) knowledge discovery from data bases; and (b) knowledge engineering and decision analysis

disciplines and processes. The proposed integration is in just one discipline and process of data and business analytics: wide knowledge discovery.

That means that the respective dual characteristics of those disciplines and processes: database–data analyst, expert–knowledge engineer, and decision maker–decision analyst are joined in just one dual process: sources–analyst by wide knowledge discovery (Figure 1) [82–85].

Decision support systems, expert systems software, and data analytics systems have advanced separately to satisfy those dual communications. Transdisciplinary approach of WKD should permit to get all together by the real achievement of AA Paradigm.

The general phase (GPh) allows principally the use of literature and expert knowledge towards general representation that should be specified by contextual sources in the particular phase (PPh). Data are not excluded as a source in GPh, because data correspondent of different contexts can inspire hypothesis and exploration in the specific context of PPh.

Consequently, following the wide knowledge discovery strategy constitutes a way towards:

1.  The accomplishment of more directly oriented to the decision-making tools,
2.  Better participation of experts and decision-makers in the analytics process, by the suggestion of hypotheses, concepts, and decision-making alternatives evaluation.

Those results are desirable and necessary for the successful evolution of data analytics towards the augmented analytics (AA) paradigm.

## 4. Scientific Strategies of Soft Computing towards Wide Knowledge Discovery

Certain scientific methodological ways can be conceptualized like useful general strategies towards wide knowledge discovery. They are described as strategies and exemplified by specific strategic results.

### 4.1. Theoretical Hybridization

Definition: Creation and use of transdisciplinary theories and models looking for desirable theoretical properties of hybrid objects. Theoretical hybridization makes emphasis on the use of theoretical considerations coming from any mathematical or computer sciences theory to look for useful hybrid systems, which can give new opportunities towards transdisciplinary approaches.

Strategic fundament: Hybridization has been a very space looking for solutions to problems. Looking for an effective way to facilitate wide knowledge discovery demands more formal approaches able to create systems and tools inspired in new transdisciplinary theories. The need for new interpretability by natural language and general integrative cognitive models is a general strategy oriented to the new paradigm of analytics.

The important role of fuzzy logic in hybridization and the growing role of mathematical fuzzy logic in soft computing is an important argument in favor of this strategy.

#### 4.1.1. Examples of Strategic Results

Compensatory Fuzzy Logic

It is a transdisciplinary theory of fuzzy multivalued logic systems, holding axioms inspired from Boolean logic, statistics as well as decision making theories and methods. The axiom which gives it the name is precisely the compensatory character of the conjunction. It is interpretable according to natural language using a new transdisciplinary concept that defines it as Interpretability or compatibility with the most important mathematical theories used to deal with social practices with natural language. It can deal with a wide spectrum of usual fuzzy logic problems, treated by different fuzzy approaches, and has been used in very different real applications case studies [9,10].

Archimedean Compensatory Fuzzy Logic

It is another transdisciplinary and pluralist theory that joins a logic of no refutation (Archimedean norm and conorm theory) and an affirmation one (compensatory fuzzy

Logic). It can deal with four different approaches by quantification with the two theories: no refutation value of possibility, no refutation value of necessity, Affirmation value of possibility and Affirmation value of necessity. It defines the compatibility of CFL logic system with an Archimedean norm and conorm system and prove they have the same ordinal behaviour [11,12]. Treatment of modifiers and membership functions like sigmoid functions and continue linguistic variables have been treated as generalizations useful for knowledge discovery that allows to get by parameters searching contextual logics useful for each knowledge discovery problem [86]

Cooperative N-Personal Game Solution by Knowledge Engineering

A new model of solution for cooperative n-personal games was created using knowledge engineering from negotiation literature and experts' consultancy. It joins propositions transformed by natural language using the logical system that some authors call probabilistic logic, which uses probabilistic sum and product as norm and conorm respectfully. It was successfully compared with the most used alternatives as solution models. Certain convergence needed was shown experimentally and the model was applied to the analysis of real negotiations [87,88]. A behavioral bargaining experiment was used towards parameters estimation. This result is a particular case of an important strategy that could be very useful as a particular form of WKD and theoretical hybridization towards transdisciplinary sciences: Knowledge discovery by knowledge engineering (KDBKE). That consists of defining concepts and making hypotheses using Fuzzy Logic Predicates literature from a practical knowledge domain. This particular result was obtained as an important element of games theory: a new solution concept of cooperative n-personal games from negotiation literature. A new model was elaborated using KDBKE, by CFL, with the intention to achieve better properties for decision making by it using the Interpretability properties of that logical theory [89–92].

*4.2. General Hybridization (GH)*

Definition: Wide hybridization using any possible useful mathematical approach
Strategic fundament: The disciplinary science division is an obstacle towards general integrative cognitive models, so important for the achievement of augmented intelligence and Augmented Analytics Paradigm. Particularly the definition of Soft computing as a discipline that works with hybridization of specific 'soft' mathematical elements, is an implicit limit used by soft computing authors. Tendency towards a more open science as well and transdisciplinary research, as well as logical and scientific pluralism are opportune elements for an important impact. The emergence of data analytics and business Analytics as disciplines are implicitly open spaces to receive new effective hybrid approaches and use them towards decision making and problem solving. the use of WKD as a scientific and technological strategic vision recommends the use of all the opportunities for hybridization. General hybridization extends the classical way of hybridization in the tight margins of soft-computing or computational intelligence towards any mathematical or computer sciences method. The protagonism of fuzzy logic hybridization and the growing role of narrow fuzzy logic in soft computing will make feasible the combination with mathematical theories with a high formalization level.

4.2.1. Examples of Strategic Results
Definition of Universal Proposition over Cartesian Products of Intervals in CFL

The universal proposition was obtained from the discrete case, using integrals and the medium value theorem. It was useful selecting the best implications to complete compensatory inference systems with the best performance by the propositional logic Kleene axioms [93,94].

Generalization of sigmoidal functions in the context of Archimedean compensatory fuzzy logic by using the correspondent differential equation.

The usual Differential equation solution used as justification for the membership sigmoid functions definition was used for generalization in the framework of Archimedean compensatory fuzzy logic. Necessary and sufficient condition theorem is proved [86].

Wide Knowledge Discovery by Fuzzy Predicates (WKDFP)

Definition: It is a strategy to get WKD. The use of hybrid knowledge discovery models using fuzzy predicates as a form of representation, CFL and ACFL as ways of evaluation, and any evolutionary, heuristic, or optimization method as methods for searching.

Strategic Fundamentals

The recognition of fuzzy sets and fuzzy logic disciplines as sciences of vagueness, their graphical interpretability, as well as the advances in natural language interpretability, especially the ones of CFL and ACFL, are the reasons for the recommendation of KDFP as a concrete strategy of advancing towards WKD. KDFP has been a very good strategy for successful applications of data and business analytics. Computational systems have been created and used as semantic integrators for analytics using this strategy [6–8,95,96]. A digital platform of business analytics by fuzzy predicates of ACFL and norm and conorm approaches have been created and used. It uses a genetic algorithm able to look for any logical structure predicates with high value of the correspondent universal propositions by a combination of the logical operators and parameters [48,49]. Its name is Eureka Universe (EU). Some examples of real applications using EU can be appreciated in the following website [97–102].

Examples and Explanation of Strategic Results

*Images and signals treatment as well as compensatory morphology:*

WKDFP have been used successfully for image treatment [103–108]. Specifically, they were created as alternatives to mathematical morphology and fuzzy morphology a new way called compensatory morphology, where the excellent robustness of CFL operators are used successfully, using new definitions of the basic morphology operators by CFL [109–111]. This is an interesting property because even when CFL is very sensitive because of the property of strict growing of its definition [9,10,112], compensatory axiom guarantees robustness because of the compensation of the small differences among different attributes [112]. That is the key to its success in treatment with the presence of measurement noise.

*SWOT-OA model for strategic management:*

A fuzzy model called SWOT-OA has been created by knowledge engineering using fuzzy predicates modeling strategic knowledge. The model allows expressing strategic changing projects as rankings of strategies or objectives [113–117].

*Knowledge Engineering and decision-making applications:*

Using knowledge engineering and decision analysis by fuzzy predicates, it is possible to incorporate expert knowledge and knowledge about preferences as parts of data and business analysis. Some models and case studies, including spatial data infrastructure readiness for different countries [118–120], protected areas management of the Valdes Peninsula in Argentina [95,96], companies sustainability report convenience and readiness in companies of Germany and Cuba [121], as well as a compensatory fuzzy logic model of technical trading [122], are some good examples of the wide range of possible applications.

These two strategies of hybridization should be applied together; the first one is led to get wider the space of hybridization, the second one is directed to get deeper into the properties and operation of the hybrid objects by the elaboration of new transdisciplinary theories by different integrated initiatives.

That joint strategy could be called transdisciplinary hybridization (Thy), that name emphasizes simultaneously the not exclusion of any mathematical theory for hybridization

(even when they have not been considered part of soft computing) and the use of theoretical approaches because the creation of new hybrid theories spaces is just the way to get transdisciplinarity.

The cross-impact matrix in Table 2, obtained from the one in Section 2, illustrates that strategies WKDFL: wide knowledge discovery by fuzzy logic predicates and THy: transdisciplinary hybridization can influence the better use of all the opportunities and threats.

**Table 2.** Cross-impact matrix with strategies.

|          | 1 TS-SLP | 2 LGCM | 3 LIDL | 4 TGG | 5 DLAR |
|----------|----------|--------|--------|-------|--------|
| 1 SciV   |          | WKDFLP | WKDFLP | WKDFLP |       |
| 2. RMFL  | THy      |        |        |       | THy    |
| 3 ATNLI  |          | WKDFLP | WKDFLP | WKDFLP |       |
| 4 GIFL   | THy      |        | THy    |       | THy    |
| 5 DEA    | THy      |        |        |       | THy    |
| 6 NARNLT |          | WKDFLP | WKDFLP | WKDFLP |       |
| 7 FCD    | THy      |        |        |       | THy    |

Figure 2 illustrates the way that strategies are connected towards the achievement of WKD. That accomplishment is obtained by wide knowledge discovery by fuzzy predicates (WKDFP), which in turn is achieved by CFL, ACFL development, and other results. One of them is cooperative games theory by knowledge engineering (CGTHKE). Those results are possible by methodological strategies like THy, TH, GH, and KDBKE.

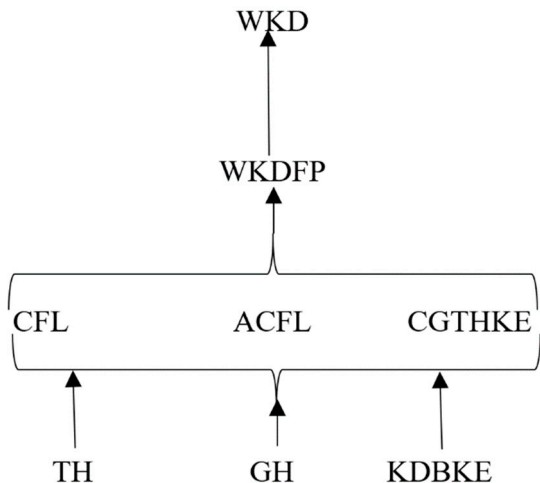

**Figure 2.** Relationships among strategies.

## 5. Related and Future Work

The three principal dimensions in progress towards wide knowledge discovery as data and business analytics based on soft computing are CFL emergence and development, ACFL creation and development, and development of a transdisciplinary approach to business analytics based on fuzzy predicates. Some future works planned are explained as well.

### 5.1. Emergence and Development of CFL (From 2005)

A new theoretical approach interpretable by bivalent logic called CFL and compatible with selected elements of decision-making theories and statistics was created [9,10,123].

Computational Tools for their use were created, and multiple applications were made in [9,10,89–92,95–111,113–122]. A methodology for knowledge management and decision making in organizations was elaborated in [6–8].

Some experiments testing the compatibility of CFL with human behavior were made, like a fuzzy way to understand experiments of Kahneman and Tversky [124,125] and a study of the best fuzzy operators in the framework of constructive decision-making approach [126]

Experiments for testing the compatibility of CFL with problems of data mining, like the discovery of association and classification rules [9,10,127], have been made successfully.

### 5.2. Creation and Development of ACFL (From 2014)

The elaboration of ACFL as an even better theoretical space towards knowledge discovery has been created as a pluralist transdisciplinary theory [9,10].

Generalizations of a broader space including modifiers, sigmoid membership functions, and continuous linguistic variables using theoretical hybridization have delivered the opportunity to advance new useful ways to do with fuzzy predicates knowledge discovery [86]

### 5.3. Developments of Business Analytics by CFL and ACFL (From 2014)

The Archimedean compensatory fuzzy logic is being used for new advances by general and theoretical hybridization, with a positive impact on analytics performed by fuzzy predicates. They are in the process of implementation into the Eureka Universe platform [97,98].

### 5.4. Future Works

Towards interpretability by natural language:

- The proposed principle of representation of linguistic variables and the correspondent algorithms for the transformation of membership functions of the general linguistic continue variable into a predefined set of labels [128,129] will be generalized to ACFL.
- Necessary and sufficient conditions theorems for a special free context grammar have been planned to be obtained from the use of generalized continue linguistic variables.
- Elaboration of semantic models for syntactic expressions based on the minimalist program

New hybridization results:

- Mixing with neural networks as a strategy to new more effective forms of representation and searching is planned. An initial neural network already implemented by using ACFL operators and generalized linguistic variables will be used as a point of departure to get advances in deep learning towards advances in fuzzy predicates analytics and neural networks interpreted by them.
- A Logical-probabilistic inference using the statistical and logical properties of compensatory fuzzy logic [9,10,93,94,130] will be developed.
- The use of new genetic algorithms and other ways of searching towards wider classes of fuzzy logic predicates should have a meaningful development and impact in interpretability and inference accuracy.

Applied models for analysis and decision making

- Advances towards compensatory morphology in the treatment of color images are planned. The pluralism of ACFL will be used to get a model by integration of different windows using specific ACFL logics.
- New models for decision making in different areas of management should be obtained, for example: internationalization of companies, social enterprises, supply chain management, human capital and competencies, etc.

### 6. Conclusions

Arguments in favor of what kind of mathematical elements should be introduced into soft computing towards a better impact on the area of data analytics have been offered.

Why and how they can be introduced, as part of the study and analysis made here, is a relevant contribution as well.

The use of wide knowledge discovery as the principal strategy towards soft computing contribution to Augmented Analytics should be effective and has shown significant advances made by general strategies like general hybridization, theoretical hybridization, and knowledge discovery by fuzzy predicates. Some results obtained by these strategies have been addressed and explained in this paper. The development of compensatory fuzzy logic and Archimedean compensatory fuzzy logic are probably the two most important strategic results in this context. The former works towards a new transdisciplinary concept of interpretability, and the latter as a good contribution to advance towards a general integrative cognitive model, necessary for augmented analytics, by advancing towards a logical and scientific pluralism.

Knowledge engineering for theoretical hybridization is a particular strategy with a great potential for transformation from fuzzy predicates, as it is very well illustrated by the cooperative games solutions created.

A program of research organized by three dimensions has been evolved from the creation of compensatory fuzzy logic towards Archimedean compensatory fuzzy logic and the study of their properties towards new advances in the building of analytics by fuzzy predicates with the relevant role of hybridization.

The advances of this plan have discovered a concrete way to achieve WKD. Such a plan uses predicates as the form of representation, CFL and ACFL logical systems as the way of evaluation, and genetic algorithms as the way of searching. This is called "WKD by fuzzy logic predicates" (WKDFLP).

Alternative representations of predicates constitute an important source of hybridization; for example, neural network representations are studied, and the correspondent use of back propagation.

The buildup of semantic integrators like the Eureka Universe platform is a sound and promising contribution towards the realization of the augmented analytics paradigm using WKDFLP.

Future works included in Section 4 should complete the study made till now. However, some elements like accuracy improvement in complex and high dimensional cases of fuzzy control and the compatibility with protoforms used in CWW and LDS have remained almost unexplored till now.

**Author Contributions:** R.A.E.-A. Conceptualization, Investigation, Methodology, Project administration, Writing—original draft, Visualization. W.P. Conceptualization, Supervision, Writing—review & editing. E.S. Data curation, Formal analysis, Resources, Writing—review & editing. L.C.-R. Validation, Visualization, Writing—review & editing. All authors have read and agreed to the published version of the manuscript.

**Funding:** This research received no external funding.

**Informed Consent Statement:** Not applicable.

**Data Availability Statement:** Data sharing not applicable.

**Conflicts of Interest:** The authors declare no conflict of interest.

## Appendix A. Three Dimensions Research Program in Course

Creation and development of CFL (From 2005)

1. Creating a new theoretical approach interpretable by bivalent logic, and compatible with selected elements of decision-making theories and statistics
2. Elaboration of computational tools
3. Applications to different fields

    3.1. Application to different fields of business and management
    3.2. Application to images and signals treatment

  3.3.  Application to other fields
4. Elaboration and application of experiments testing the following elements:
  4.1.  Compatibility with human behavior
  4.2.  Compatibility with knowledge discovery methods for particular problems
  4.3.  Compatibility with Mamdani approach in simple cases
  4.4.  Performance of different deductive structures as heuristics in knowledge discovery as a way of approximate reasoning
  4.5.  Accuracy improvement in complex and high dimensional cases of fuzzy control
  4.6.  Compatibility with protoforms used in CWW and LDS
  4.7.  Compatibility with utility theory under risk and prospect theory
  4.8.  Study of operators robustness

Creation and development of ACFL (From 2014)

1. Elaboration of a pluralist theoretical approach for fuzzy logic which allows the values compatibility between CFL and the usual very common and extended approach of Fuzzy Logic, the norm and co-norm approach.
2. Elaboration of generalizations of important concepts of fuzzy logic, which can be associated by parameters to different ACFL systems with the purpose to include them as searching parameters for knowledge discovery, as way of contextual pluralism.
  2.1.  Generalization of modifiers, sigmoid membership functions and linguistic continue variables for each logic.
  2.2.  Study of negations, implications, equivalence, and similarities in ACFL.
  2.3.  Elaborate interpretations of exigence level and risk attitude from Archimedean compensatory fuzzy logic. as way of achievement of individual and groups pluralism.
  2.4.  Elaborate formulas for dual, intuitionistic, and neutrosophic fuzzy logic using archimedean compensatory fuzzy logic as point of departure.

Developments of Business Analytics by CFL and ACFL (From 2014)

1. Creation and development of Eureka Universe.
2. Elaboration of procedures for the association of fuzzy predicates forms from universal grammar syntactical structures.
3. Elaboration of a way of representation of fuzzy predicates by neural networks which could join the interpretability properties of CFL to the use of the extraordinary results of deep learning in data analysis by fuzzy predicates.
4. Elaboration of models and cognitive flows for relevant decision-making problems in business.
5. Elaborate an ACFL games theory-based approach useful for business analysis and negotiation.
6. Development of Hybridization with other relevant mathematical methods.
  6.1.  Hybridization with simulation
  6.2.  Hybridization with constructive decision-making approaches
  6.3.  Development of fuzzy prospect theory by ACFL
  6.4.  Hybridization with statistical and stochastic approaches
  6.5.  Hybridization with optimization and evolutionary searching
7. Development of logic-statistical inference methods by CFL and ACFL.
8. Development of the Generalized Linguistic Continue Variables as a Free Context Grammar theoretically developed from ACFL results.
9. Implementation of the new elements in Eureka Universe.

## Appendix B. List of Acronyms

i. Advances towards interpretability by natural language (ATNLI)
ii. Augmented analytics (AA)
iii. Archimedean compensatory fuzzy logic (ACFL)

iv.   Compensatory fuzzy logic (CFL)
v.    Computing with words (CWW)
vi.   Deep learning extraordinary accuracy results (DLAR)
vii.  Development of evolutionary algorithms and metaheuristics (DEA)
viii. Education 4.0 (E4.0)
ix.   Eureka Universe (EU)
x.    Existence of universal transformational generative grammar, a consensual scientific linguistic model (TGG)
xi.   Fuzzy logic as the science of vagueness (SciV)
xii.  Fuzzy control deviation towards functional adjustment techniques (FCD)
xiii. General phase (GPh)
xiv.  Graphical interpretability by trees, graphs, and networks (GIFL)

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
