# Peer review of "Transdisciplinary Scientific Strategies for Soft Computing Development: Towards an Era of Data and Business Analytics"

_axioms, doi:10.3390/axioms10020093_

Round 1

Reviewer 1 Report

In this manuscript, the authors present the characteristics of contemporary sub-areas of soft computing (compensatory fuzzy logic and Archimedean fuzzy logic) and fuzzy predicates as tools for knowledge discovery.

Abstract:

Please, edit the text to avoid the repetitions of “strategic” and “strategies”.

Keywords:

"Metaheuristics" could be excluded.

Introduction:

The main contribution is not clear. The link to this journal is not ensured.

Please clarify the meaning of “strategy” and “Master strategy” and then describe the characteristics of strategies for soft computing development.

In my opinion, a new figure could be added to visualize the relationships between Computational Intelligence, Soft Computing, Data Mining, Data Analytics, Business Analytics, Augmented Analytics and Hybrid Augmented Analytics.

p. 2-3: Please, do not use the terms ‘strategy’ and ‘paradigm’ interchangeably (p. 2: Augmented Analytics strategy, p. 3: Augmented Analytics (AA) Paradigm).

p. 2: “It multiplies the importance … “ – In this sentence, the subject is unclear.

The Edurecas Community role is not clarified.

I. SWOT Analysis of Computational Intelligence towards Data and Business Analytics:

The relationship Computational Intelligence – Soft Computing should be clarified.

“However, those advances …” – This sentence is too long.

 “That has produced …” – In this sentence, the subject is unclear.

II. Wide Knowledge Discovery strategy towards AA Paradigm: Philosophical, Cognitive and strategic arguments:

Figure 1: The abilities of WKD to represent, evaluate and search knowledge should be visualised. Two phases (Particular Phase and General Phase) should be commented on.

p. 7: “it is a …” – In this sentence, the subject is unclear.

p. 7: “That means that the respective dual characteristics of those disciplines and processes: Data Base – Data Analyst, Expert – Knowledge Engineer, and Decisor – Decision Analyst are joined in just one dual process: Sources – Analyst by Wide Knowledge Discovery (Figure 1) [71-74].” – Please, edit this fragment. Data Analyst, Expert, Knowledge Engineer, Decision Analyst are actually IT users and business users.

p. 7: “Decisor” -> “Decision Maker”.

III. General scientific strategies of soft computing towards Wide Knowledge Discovery:

According to Section II, Wide Knowledge Discovery is a strategy. TH, GH and KDFP are strategies for Wide Knowledge Discovery in section III. According to Appendix 1, there are one principal, four general and six others strategies. – Please, discuss their relationships.

p. 9: “KDBKE” – The abbreviation is undefined.

p. 9: “dominium” -> “domain”

p. 10: “factible” – Please, change this word.

IV. Related and future work:

Future work should be moved in Conclusion section.

V. Conclusions and Recommendations:

This section should be rewritten by native English editor.

p. 14: “Arquimedean” – “Archimedean”

References:

This section should be reformatted according the journal requirements. There are three references #69 and two references #70. Please, edit the references list to avoid the repetitions.

Appendix 1:

The table caption is missing.  The table content should be discussed.

Reviewer 2 Report

Paper deals with important task. Authors have provided a review and analysis paper that offers a strategic vision for Soft Computing development towards a wider favorable impact in Data Analytics

Paper has practical value.

It has a logical structure.

Suggestions:

  1. Abstract section should be extended usinf motivation of this paper as well as results obtained
  2. It would be good to add point-by-point the main contributions in the end of the Introduction section
  3. It would be good to add the reminder of this paper
  4. Related works section should be added to the paper
  5. Conclusion section should be extended using: 1) results obtained in the paper; 2) limitations of the conducted study
  6. Some references are outdated. Please fix it using 3-5 years old papers in high-impact journals.

Other suggestions

  1. Paper should be formatted according to the Journal rules

Reviewer 3 Report

 Transdisciplinary scientific strategies for Soft Computing development: towards an era of Data and Business Analytics

The authors presented a review paper concerningstrategic vision for Soft Computing development towards a wider favorable impact in Data Analytics”. The authors defined and illustrated strategies. The SWOT Analysis of Computational Intelligence towards Data and Business Analytics is presented. The authors enough clearly describe their original contribution in this paper.  The goals of the paper are properly defined. The introduction clearly states the problem being investigated but the main part of the paper needs improvement.  The choice of references is quite satisfactory.

I found this work interesting and introducing a new impact in the systemization and methodological discussion about Soft Computing development. Yet, I have a few remarks which could be taken into consideration to improve the quality of the paper.  

  1. Firstly, some technical remark. The paper does not fit mpdi journal formatting requirements. The editing form of the manuscript is very messy (see font, size, type, references, style), which negatively affects the substantive perception of the paper.
  2. Appendix 1: Strategic elements and SWOT Matrix (of what?). The title is imprecise
  3. Please, systemize and ordering the form of references. For instance, the title is missing
  4. Norma P. Rodríguez-Cándido, Rafael A. Espin-Andrade, Efrain Solares and Witold Pedrycz Axioms 2021, 10(1), 36; doi:10.3390/axioms10010036
  5. Citation “The emergence of fuzzy sets and fuzzy logic theories is celebrated commemorating the date of Lofti’s first paper publication” ( The family name Zadeh is is missing in a sentence as well citation his papers)
  6. I proposed in the part: SWOT Analysis of Computational Intelligence towards Data and Business Analytics separately described: strengths, weaknesses, opportunities, and threats of Soft computing.
  7. Figure 1. Transdisciplinary Data Analytics and WKD is too general and do not give any useful information. Please remove it or give a more detailed description or specification.

In my opinion, to be consistent with the previous presentation SWOT analysis Appendix 1 should be introduced to the main text with a more detailed description.

  1. The title in a table in the Appendix is missing. The strategies should be more exactly described.

Conclusions.  I recommended accepting this paper with the proposed revision.

Round 2

Reviewer 1 Report

Regardless of the revision made, the manuscript still does not meet the requirements for publication in a high-quality journal such as MDPI Axioms. Please, read and edit the whole text to avoid the inconsistencies.

Abstract: Please, edit the text to clarify the main authors’ contributions.

In order to improve the readability, please include a list of all abbreviations.

Some technical remarks:

p. 7: The table caption is missing.

p. 9: Figure 1 is trivial and should be redrawn with more details or omitted.

The Table 1 and Table 2 should be mentioned in the manuscript’s text.

--

p. 11: “A detailed list of strengths, weaknesses, opportunities, threats, and strategies, as well as a Cross Impact Matrix, are included in Appendix 1.”

and

p. 16: “A scientific program with past, present, and future works, including those three dimensions, is more detailed in Appendix 2.”

Please, edit these sentences. Appendix 1 and Appendix 2 do not exist.

--

p. 15: “One of them is Cooperative Games Theory by Knowledge Engineering.” -> “One of them is Cooperative Games Theory by Knowledge Engineering (CGTHKE).”

p. 15: The term “WKDFP” should be defined here, instead at the end of manuscript (p. 18).

The manuscript should be reformatted according the journal requirements.

Reviewer 2 Report

Paper should be formated according to the Journal rules!

Reviewer 3 Report

The authors have improved the manuscript a lot. Most of my remarks have been taken into account. But the paper does not fit midi journal formatting requirements. I can repeat my remark: “The editing form of the manuscript is very messy (see font, size, type, references, style), which negatively affects the substantive perception of the paper”.  Moreover, the references are not formatting properly. In my opinion, properly formatting paper could be ready for publication.  I do not need to review the manuscript more.
